# Prostaglandin F2α Affects the Cycle of Clock Gene Expression and Mouse Behavior

**DOI:** 10.3390/ijms25031841

**Published:** 2024-02-02

**Authors:** Yuya Tsurudome, Yuya Yoshida, Kengo Hamamura, Takashi Ogino, Sai Yasukochi, Shinobu Yasuo, Ayaka Iwamoto, Tatsuya Yoshihara, Tomoaki Inazumi, Soken Tsuchiya, Toru Takeo, Naomi Nakagata, Shigekazu Higuchi, Yukihiko Sugimoto, Akito Tsuruta, Satoru Koyanagi, Naoya Matsunaga, Shigehiro Ohdo

**Affiliations:** 1Department of Pharmaceutics, Faculty of Pharmaceutical Sciences, Kyushu University, 3-1-1 Maidashi, Higashi-ku, Fukuoka 812-8582, Japan; tsurudome-y19@rs.socu.ac.jp (Y.T.); ogino.takashi.029@m.kyushu-u.ac.jp (T.O.); yasukochi-s16@umin.ac.jp (S.Y.); koyanagi@phar.kyushu-u.ac.jp (S.K.); 2Department of Clinical Pharmacokinetics, Faculty of Pharmaceutical Sciences, Kyushu University, 3-1-1 Maidashi, Higashi-ku, Fukuoka 812-8582, Japan; yoshida@phar.kyushu-u.ac.jp (Y.Y.); hamamura@phar.kyushu-u.ac.jp (K.H.); 3Regulation in Metabolism and Behavior, Faculty of Agriculture, Kyushu University, 744 Motooka, Nishi-ku, Fukuoka 819-0395, Japan; syasuo@brs.kyushu-u.ac.jp (S.Y.);; 4SOUSEIKAI Fukuoka Mirai Hospital Clinical Research Center, 3-5-1 Kashiiteriha, Higashi-ku, Fukuoka 813-0017, Japan; tatsuya-yoshihara@lta-med.com; 5Department of Pharmaceutical Biochemistry, Graduate School of Pharmaceutical Sciences, Kumamoto University, 5-1, Oe-Honmachi, Chuo-ku, Kumamoto 862-0973, Japan; tinazumi@kumamoto-u.ac.jp (T.I.); sokent@kumamoto-u.ac.jp (S.T.); ysugi@kumamoto-u.ac.jp (Y.S.); 6Division of Reproductive Engineering, Center for Animal Resources and Development (CARD), Institute of Resource Development and Analysis, Kumamoto University, 2-2-1 Honjo, Chuo-ku, Kumamoto 860-0811, Japan; takeo@kumamoto-u.ac.jp; 7Division of Reproductive Biotechnology and Innovation, Center for Animal Resources and Development (CARD), Institute of Resource Development and Analysis, Kumamoto University, 2-2-1 Honjo, Chuo-ku, Kumamoto 860-0811, Japan; nakagata@kumamoto-u.ac.jp; 8Department of Human Life Design and Science, Faculty of Design, Kyushu University, 4-9-1 Shiobaru, Minami-ku, Fukuoka 815-8540, Japan; higu-s@design.kyushu-u.ac.jp; 9Department of Glocal Healthcare Science, Faculty of Pharmaceutical Sciences, Kyushu University, 3-1-1 Maidashi, Higashi-ku, Fukuoka 812-8582, Japan; tsuruta@phar.kyushu-u.ac.jp

**Keywords:** mouse behavior, prostaglandin F_2α_, circadian clock, locomotor activity, suprachiasmatic nucleus

## Abstract

Prostaglandins are bioactive compounds, and the activation of their receptors affects the expression of clock genes. However, the prostaglandin F receptor (*Ptgfr*) has no known relationship with biological rhythms. Here, we first measured the locomotor period lengths of *Ptgfr-KO* (B6.129-*Ptgfr^tm1Sna^*) mice and found that they were longer under constant dark conditions (DD) than those of wild-type (C57BL/6J) mice. We then investigated the clock gene patterns within the suprachiasmatic nucleus in *Ptgfr-KO* mice under DD and observed a decrease in the expression of the clock gene cryptochrome 1 (*Cry1*), which is related to the circadian cycle. Moreover, the expression of *Cry1*, *Cry2*, and *Period2* (*Per2*) mRNA were significantly altered in the mouse liver in *Ptgfr-KO* mice under DD. In the wild-type mouse, the plasma prostaglandin F_2α_ (PGF_2α_) levels showed a circadian rhythm under a 12 h cycle of light–dark conditions. In addition, in vitro experiments showed that the addition of PTGFR agonists altered the amplitude of *Per2*::luc activity, and this alteration differed with the timing of the agonist addition. These results lead us to hypothesize that the plasma rhythm of PGF_2α_ is important for driving clock genes, thus suggesting the involvement of PGF_2α_- and *Ptgfr*-targeting drugs in the biological clock cycle.

## 1. Introduction

Various physiological and biological functions show a 24 h rhythmic cycle in synchronization with the external light environment. Periodic fluctuations are driven by an endogenous molecular system, known as the circadian clock [1]. The core of the circadian clock constitutes a transcription–translation feedback loop. Heterodimers consisting of CLOCK and BMAL1 promote the transcription of period (PER) and cryptochrome (CRY) genes by binding to E-box elements [2,3]. Upon reaching critical amounts, PER and CRY proteins dampen transcriptional activation via CLOCK/BMAL1, thereby generating circadian oscillations in their own transcription [4]. Clock genes, responsible for core oscillation loops, regulate the circadian expressions of clock-controlled output genes, such as albumin D-site binding protein (DBP), E4BP4 (NFIL3), and peroxisome proliferator-activated receptor-α (PPARα) [5]. DBP and E4BP4 regulate circadian gene expression by competitively binding to the D-site of the same DNA sequence [6]. Likewise, PPAR response elements (PPREs) are required for the circadian expression of PPARα target genes [7]. These mechanisms ultimately regulate various downstream events in the transcription, translation, and degradation processes [8,9].

The suprachiasmatic nucleus (SCN) of the brain is the master circadian clock that synchronizes the circadian clock mechanisms of individual cells [10]. Neurally active signals associated with photoreception in the retina activate the cAMP response element-binding protein (CREB) in the SCN, which, in turn, activates the transcription of the *Per* gene [11]. Increased *Per* expression triggers the transcription rhythm of clock genes in the SCN, evoking glucocorticoid secretion and sympathetic activation [12]. Secreted hormones and cytokines synchronize the clock genes of each cell in the peripheral organs under an external light environment [13].

Prostaglandins are endogenous, biologically active substances synthesized via arachidonic acid. Receptors corresponding to each prostaglandin are expressed in organs throughout the body, performing numerous physiological activities by mediating downstream signaling [14]. The plasma concentrations of some prostaglandins follow a 24 h cycle [15]. Although they are known to affect the expression of clock genes [16], no prostaglandins have been reported to affect the periodicity of clock genes or mouse circadian behavior.

Prostaglandin F_2α_ (PGF_2α_) is a subtype of prostaglandin that acts on blood vessels around the glomus body and retinal arteries [17]. The prostaglandin F receptor (PTGFR) is localized in the eye and uterus, where its ligand binding activates PKC and AMPK [18]. Latanoprost is a highly selective agonist of PTGFR and a strong PTGFR agonist with weak prostaglandin E receptor effects [19]. Whether PTGFR signaling and latanoprost affect the expression cycle of clock genes or mouse behavior remains unclear. Here, we aimed to clarify the relationship between PGF_2α_ and the circadian clock mechanism by analyzing the locomotor activity rhythm and expression of clock genes in *Ptgfr-KO* mice.

## 2. Results

### 2.1. Ptgfr-KO Mice Exhibit Longer Behavioral Cycles under Constant Dark Conditions

We initially observed differences in the behavioral cycles, meaning the time length between the start of an activity and the start of the next activity, of wild-type and *Ptgfr-KO* mice under constant dark (DD) conditions. After acclimation to light–dark (LD) conditions for two weeks, each mouse was kept in DD conditions. Wild-type mice exhibited a behavioral cycle of 23.58 ± 0.293 h (Figure 1a), similar to those in previous reports [12]. In contrast, *Ptgfr-KO* mice exhibited a longer behavioral cycle (24.05 ± 0.085 h) than wild-type mice (Figure 1b). These results indicate that Ptgfr deficiency affects the periodicity of the biological circadian clock.

### 2.2. Time-Dependent Changes in Expression of Clock Genes in the Suprachiasmatic Nucleus (SCN) of Ptgfr-KO Mice

Changes in the amplitude of clock gene expression in the SCN affect the periodicity of behavior [20]. Because the locomotor activity cycle of *Ptgfr-KO* mice was longer than that of wild-type mice, we evaluated the expression rhythms of clock genes in the SCN of *Ptgfr-KO* mice. We measured the expression rhythms of *Per1*, *Per2*, *Bmal1*, and *Cry1*, which form the core of the circadian clock system, and whose own expression indicates circadian oscillations [21], using in situ hybridization. The results show that the mRNA expression of *Per1*, *Per2*, and *Bmal1* was nearly equal in both wild-type and *Ptgfr-KO* mice (Figure 2a,b,d). In contrast, the peak time of *Cry1* mRNA expression was regressed in *Ptgfr-KO* mice (Figure 2c; Appendix A: *Cry1* in wild-type; acrophase = 8.99, *Cry1* in *Ptgfr-KO*; acrophase = 13.60). In addition, the expression level at the peak time was also significantly downregulated. These results indicate that *Ptgfr* deficiency affects time-dependent *Cry1* expression in SCN.

### 2.3. Time-Dependent Changes in Clock Gene Expression in the Liver of Ptgfr-KO Mice

The altered expression of clock genes in the SCN also alters their expression rhythms in peripheral organs, such as the liver and kidneys [10,11]. When changes are observed in the clock core–loop rhythms of peripheral organs, the expression of clock-regulated genes undergoes rhythmic changes [6,7]. Because changes in clock genes were observed in the SCN of *Ptgfr-KO* mice, we measured their expression levels in the liver. Total RNA was extracted from the livers of wild-type and *Ptgfr-KO* mice after 3 days under DD conditions and reverse-transcribed. The mRNA for clock genes was measured using a reverse transcription qualitative PCR (RT-qPCR) system. In wild-type mice, significant circadian rhythms were observed in the expression of each clock gene (Figure 3a–f). *Ptgfr-KO* mice also showed a significant circadian rhythm; however, the amplitude of *Per2* mRNA expression was reduced in *Ptgfr-KO* mice compared to that in wild-type mice (Figure 3b; Appendix A). The expression amplitudes of *Cry1* and *Cry2* mRNAs in *Ptgfr-KO* mice were higher than those in wild-type mice (Figure 3d,e; Appendix A: *Cry1* in wild-type; 0.424, *Cry1* in *Ptgfr-KO*; 0.732, *Cry2* in wild-type; 0.193, *Cry2* in *Ptgfr-KO*; 0.424). In addition, the peak time of the *Cry1/2* mRNA expression came faster. These results indicate that *Ptgfr* deficiency affects time-dependent clock gene expression in the peripheral organs.

### 2.4. The Expression Rhythms of Ptgfr and the Secretion Rhythms of PGF_2α_ in Mouse Retina

Although *Ptgfr* is expressed in organs throughout the body, it is highly expressed in the uterus and retina [17,22]. Circadian rhythms in prostaglandin E_2_ secretion in serum and prostaglandin D synthase expression in the brain have also been observed [16,23,24]. To clarify the activation rhythm of the *Ptgfr* cascade in the retina, we measured the expression rhythm of *Ptgfr* in the retinas of wild-type mice using an RT-qPCR system. The retinal *Ptgfr* mRNA expression showed a significant circadian rhythm, increasing during the dark period (Figure 4a). The secretory rhythm of PGF_2α_ in mouse serum and aqueous humor was measured using a PGF_2α_ ELISA kit. Serum PGF_2α_ levels did not exhibit a significant circadian rhythm (Figure 4b). In contrast, the PGF_2α_ concentrations in the aqueous humor exhibited the same phase rhythm as the expression rhythm of *Ptgfr* in the retinas of wild-type mice (Figure 4c). Based on these results, the PTGFR signal in the retina is considered high in the early dark period. Given the expression rhythm of *Cry1* in the SCN of wild-type and *Ptgfr-KO* mice, PTGFR signaling in the early dark period may be involved in the *Cry1* expression cycle.

### 2.5. Effects of PTGFR Agonists on Per2 Expression Rhythm in Per2::luc C6 Cells

The circadian rhythms of *Ptgfr* expression and PGF_2α_ secretion were observed, suggesting that PGF_2α_-related signaling may have a time-dependent effect. To determine the effect of PGF_2α_ signaling on the expression of clock genes, we evaluated the effect of PTGFR agonists in a model in which the clock genes were synchronized. Using *Per2*-fusion luciferase-expressing C6 rat glioma cells (*Per2*::luc C6 cells), we synchronized cell clock genes with dexamethasone, added PTGFR agonists at different times, and tested their effects on the periodicity of *Per2*::luc activity. After the *Per2*::luc C6 cells synchronized, the rhythms of *Ptgfr* and *Per2* mRNA expression were measured.

A significant rhythm in the expression of *Ptgfr* and *Per2* mRNA was observed in *Per2*::luc C6 cells (Figure 5a,b). Therefore, we added latanoprost, an PTGFR agonist, to CT0, 6, 12, and 18 and observed periodic changes in *Per2*::luc activity. A transient upregulation in *Per2* expression was observed immediately after latanoprost administration at all time points (Figure 5c–f). The results show that adding latanoprost during periods of high *Ptgfr* and *Per2* expression significantly prolonged the *Per2*::luc activity (Figure 5g). These results suggest that PTGFR signaling affects the expression rhythm of clock genes in a time-dependent manner.

## 3. Discussion

In this study, we investigated changes in the biological clocks of *Ptgfr-KO* mice. *Ptgfr-KO* mice showed a prolonged behavioral cycle of approximately 24 h. The measurement of the clock gene expression in the SCN and peripheral organs revealed changes in the expression levels and peak time of core clock genes. In particular, the expression level of *Per2* at the peak time was attenuated in the livers of *Ptgfr-KO* mice compared to that in wild-type mice, whereas the peak-time expression of *Cry1* and *Cry2* mRNA in *Ptgfr*-KO mice was higher than that in wild-type mice. Increased *Per2* mRNA expression and cycle prolongation were observed when PTGFR agonists were added. These results suggest that activating or inhibiting PTGFR may affect the expression cycle of clock genes.

Several types of prostaglandins affect clock gene expression in peripheral organs. The treatment of cells with prostaglandin E increases *Per1* expression [16]. The administration of prostaglandin E alters the expression cycle of clock genes in mice [16]. The amplitude of clock gene expression is modified in prostaglandin D synthase KO mice [24]. However, administering prostaglandin E has no effect on the behavioral cycle [16]. Furthermore, in prostaglandin D synthase knockout mice, the biological clock cycle was unchanged from that of wild-type mice under DD conditions [24]. In *Ptgfr-KO* mice, the behavioral cycle was changed to 24 h under DD conditions (Figure 1). This result indicates that *Ptgfr* is the only prostanoid receptor involved in the behavioral cycle.

Although *Ptgfr* mRNA expression was not detected in the SCN, a regression in peak time and decreased expression at the peak time of *Cry1* mRNA were observed within the SCN of *Ptgfr-KO* mice. *Ptgfr* is highly expressed in the retina of mammals [16]. Downstream PTGFR signaling, as well as that of photoreceptors, is mediated by PKC or PKG activation and Ca^2+^ signaling [18,25]. The activation of retinal Ca^2+^ signaling activates the optic nerve, causing increased CREB phosphorylation or c-Fos mRNA expression in the SCN [26,27]. Therefore, the activation of PTGFR signaling projects is likely to occur in the SCN via the optic nerve. In *Ptgfr-KO* mice, the periodicity of *Cry1* mRNA expression in the SCN is maintained but its phase is regressed (Figure 2). *Cry1-KO* mice, in which the circadian cycle of *Cry1* expression is completely lost, show a shortened behavioral cycle and hastening of behavioral phases [28]. As the rhythm of *Cry1* mRNA expression is observed in *Ptgfr-KO* mice, it is unlikely that the same shortening of the behavioral cycle phase occurs in *Ptgfr-KO* mice as in *Cry1-KO* mice. The manipulation of the *Cry1* expression cycle causes a regression in the circadian clock cycle [29]. Therefore, the regression in the *Cry1* peak observed in *Ptgfr-KO* mice may be responsible for the regression in the behavioral cycle.

PTGFR not only activates Ca^2+^ signaling but also drives other physiological pathways in the livers of mice. The activation of Ptgfr in the liver accelerates glycogenesis via the CaMKIIg/p38/FOXO1 pathway [30]. Moreover, the hepatic accumulation of triglycerides is reduced with PTGFR activation. PTGFR signaling also inhibits adipose tissue differentiation by suppressing the PPARγ function [31]. The activation of these signals and pathways may affect the expression of respective clock genes by acting on their promoter regions. The transcriptional rhythm is created by the binding region of phosphorylated CREB, which is associated with Ca^2+^ signaling upstream of the *Per2* gene [32]. The reduced expression amplitude of *Per2,* as shown in Figure 3b, may have been due to the loss of *Ptgfr*, which caused a reduction in Ca^2+^ signaling. The different effects of *Ptgfr-KO* on clock gene expression in the SCN and liver may be due to this peripheral-organ-specific function of PTGFR.

In this study, periodic changes in clock genes were evaluated in the SCN and liver. The liver clock mechanism is largely affected by changes in the SCN [33]. In addition, the liver is a relatively homogeneous peripheral organ, with most of its constituent cells being hepatic parenchymal cells [34]. Therefore, in many studies, the SCN is used as the central clock and the liver as the peripheral clock as the index for evaluation [5,35,36]. Because these results suggest that the changes in clock gene expression observed in the liver of *Ptgfr-KO* mice may also be observed in other organs in the periphery, further analysis of the relationship between PTGFR and the function of peripheral organs, including the liver, will help us understand the circadian clock mechanism in the periphery.

Latanoprost, a PTGFR agonist, is used to treat glaucoma. Although the mechanism is not clearly understood, latanoprost is thought to drive IP3/Ca^2+^ signaling in retinal cells and promote aqueous humor drainage [37]. Latanoprost is one of the PGF_2α_ ligands that potentiates intracellular Ca^2+^ concentrations in in vitro experiments [38]. As shown in Figure 5, the transient increase in the transcriptional activity of *Per2* after the addition of latanoprost is thought to be due to the influx of Ca^2+^. The *Per2* cycle changes induced by the addition of latanoprost were time-dependent (Figure 5c–f). This is similar to the phase changes induced by light irradiation in mice [39]. Ca^2+^ signaling is also involved in light-irradiation-induced SCN activation and increases *Per* gene expression [11,32]. These findings suggest that the *Ptgfr* expression rhythm may be involved in Ca^2+^-signaling-induced time-dependent cyclic changes.

The central clock, which determines the overall internal clock of the body, is affected by light, physical exercise, and feeding. The peripheral clock is also driven by variations in the central clock and zeitgeber [40]. When the factors affecting the central and peripheral clocks differ from normal, the overall biological circadian clock system can modulate. The modulation of the biological clock can lead to obesity, diabetes, cardiovascular diseases, cancer, and other diseases [9,41,42,43]. Therefore, the genes that influence the biological clock and behavioral cycle have been investigated through studies focusing on several types of genes [8,44,45]. Molecules associated with the inflammatory response affect the circadian clock mechanism through various pathways [38]. Although a link between prostanoids, which are closely associated with inflammatory processes, and clock genes has been suggested, previous studies have not identified the prostaglandin receptors that initiate changes in the behavioral cycle [16,23]. The results regarding the behavioral cycle and clock gene analyses of *Ptgfr-KO* mice in this study indicate that PTGFR activation is involved in the biological circadian clock system. Various prostaglandin subtypes have different effects on clock genes. It may be possible to develop a medicine that regulates the biological clock by conducting a detailed analysis in the future.

## 4. Materials and Methods

### 4.1. Animal Experiments

*Ptgfr-KO* male mice (B6.129-*Ptgfr^tm1Sna^*, CARD ID: 2078) with C57BL/6J background and wild-type C57BL/6J male mice were maintained under a standard light–dark cycle (lights on at zeitgeber time [ZT] 0, off at ZT12) in a temperature (24 ± 1 °C)- and humidity (60 ± 5%)-controlled room with food and water ad libitum. The animals were acclimated to the LD cycle for two weeks before the experiments. There were no changes in water consumption or food intake and no significant differences in body weight between wild-type and *Ptgfr*-KO mice (Appendix A). All experiments were conducted according to the protocol approved by the Internal Committee for Animal Experiments at Kyushu University (ID: A20-133-0). All animal experiments complied with the ARRIVE guidelines and were performed in accordance with the National Institutes of Health Guide for the Care and Use of Laboratory Animals (NIH Publication No. 8023; revised 1978).

### 4.2. Measurement of Locomotor Activity Rhythm

To measure locomotor activity, the cages were placed in an infrared ray area sensor (Neuroscience, Tokyo, Japan), and locomotor activity was measured under a 12 h LD cycle for 7 days and then under a DD cycle for 7 days. The period and onset of activity under the DD cycle were determined via Cosinor analysis using Clock Lab software (Version 2.36, Actimetrics, Evanston, IL, USA) by a chronobiology expert in a blinded manner.

### 4.3. In Situ Hybridization

Wild-type and *Ptgfr-KO* mice were euthanized via cervical dislocation following isoflurane anesthesia. At six time points every 4 h starting CT 2, the brain of each mouse was harvested and flash-frozen. Slices containing SCN were prepared from the frozen brains [46]. The expression levels of clock genes in the SCN were evaluated using in situ hybridization with reference to previous studies. Antisense 45-mer oligonucleotide probes (mBmal1:1755–1799 of AB015203; mPer1:3239–3283 of AB002108; mPer2:242–286 of AF035830; and mCry1:1742–1786 of AY034432) were labeled with [35S] dATP (New England Nuclear, Boston, MA, USA) using terminal deoxyribonucleotide transferase (GIBCO-BRL, Waltham, MA, USA).

### 4.4. Quantitative RT-PCR Analysis

RNA was extracted from mouse livers using RNAiso (Takara Bio, Osaka, Japan). Reverse transcription of RNA to cDNA was performed using a Rever Tra Ace quantitative real-time PCR kit (Toyobo, Osaka, Japan). For quantitative real-time PCR reactions, THUN-DERBIRD SYBR qPCR Mix (Toyobo) and LightCycler^®^ 96 System (Roche, Basel, Switzerland) were used, and the mRNA expression levels of each gene were measured via the calibration curve method. The expression level of each gene was corrected in reference to the β-actin mRNA expression level. The PCR primer sequences are listed in Appendix A.

### 4.5. Quantification of PGF_2α_ Using ELISA

Plasma and aqueous humor samples from wild-type mice were collected at seven time points (ZT2, 6, 10, 14, 18, 22, and ZT2 + 24 h). PGF_2α_ concentrations in the collected samples were quantified using a PGF_2α_ ELISA kit (ADI-900-069; Enzo Life Sciences, Farmingdale, NY, USA). After the antibody reaction, the PGF_2α_ concentration was observed at a wavelength of 405 nm and calculated using the method described in the kit protocol.

### 4.6. Real-Time Monitoring of Circadian Bioluminescence

*Per2*-fusion luciferase-expressing C6 rat glioma cells (*Per2*::luc C6 cells) were maintained in Dulbecco’s Modified Eagle’s Medium (DMEM) supplemented with 10% fetal bovine serum (FBS) at 37 °C in a 5% CO_2_ humidified atmosphere. *Per2*::luc C6 cells were seeded in 35 mm culture dishes. After treatment with 100 μM DEX for 2 h, the medium was exchanged for DMEM with HEPES supplemented with 100 nM luciferin and 5% FBS. After wrapping the culture dishes with Parafilm, they were placed in a lumi-cycle and the intensity of *Per2*::luc was observed. After 48 h of DEX shock, latanoprost (0.005%, 50 μL) was added every 6 h (CT 0: 30 h, CT 6: 36 h, CT 12: 42 h, CT 18: 48 h). An equal volume of saline was administered to the control group. The amplitude of bioluminescence derived from *Per2*::luc was calculated 24 h after the addition of the reagent using Lumicycle analysis software (Version 2.04, Actimetrics), as previously reported [47].

### 4.7. Statistical Analysis

Statistical analyses were performed using the GraphPad Prism software (ver. 8; GraphPad Software, San Diego, CA, USA). The significance of the differences among the groups was analyzed using analysis of variance (ANOVA), followed by Tukey’s post hoc test. Differences between the groups were analyzed using two-way ANOVA, followed by Sidak’s post hoc test. Statistical significance was set at *p* < 0.05. Cosinor analyses were performed using Circadian Rhythm Laboratory Software accessed on 20 December 2023 (https://www.circadian.org/). *p* value, which were calculated from comparisons of the residuals before and after fitting, being less than 0.05 indicated the detection of a rhythm. Using this analysis, we also calculated the acrophase, the time of the peak value in the fitted cosine function. The values calculated via this analysis are shown in Appendix A. Although no statistical methods were used to predetermine sample sizes, our sample sizes were similar to those reported in previous studies [12,30,46]. The experiments were not randomized.

## Figures and Tables

**Figure 1 ijms-25-01841-f001:**
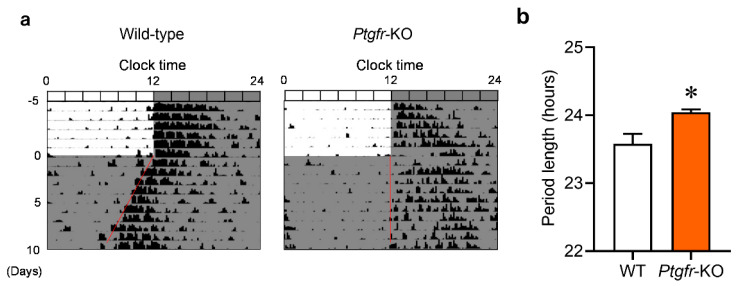
Comparison of behavioral rhythms between wild-type and *Ptgfr-KO* mice under constant dark (DD) conditions. (**a**) Representative activity records from animals initially held in a 12:12 LD cycle and then transferred to DD conditions (left: wild-type mice, right: *Ptgfr-KO* mice). (**b**) Periodogram estimates of the period for wild-type and *Ptgfr-KO* mice. Each value represents the mean with standard error; (*n* = 4). * *p* < 0.05 significant difference from wild-type mice, *t*-test.

**Figure 2 ijms-25-01841-f002:**
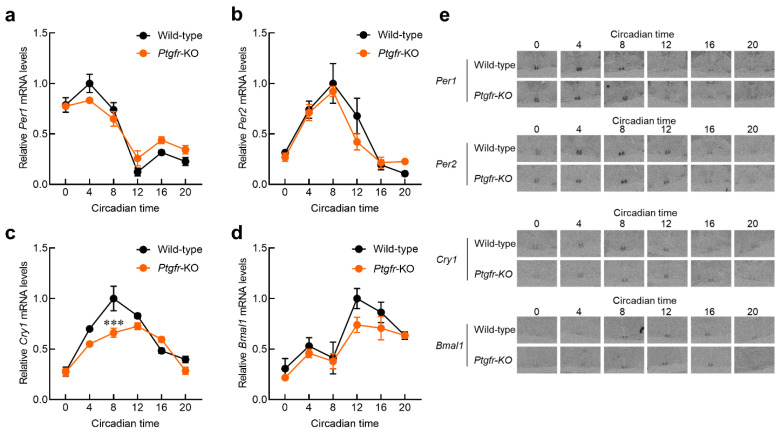
Influence on the expression of clock genes in the SCN of wild-type and *Ptgfr-KO* mice. (**a**–**d**) Temporal expression profiles of *Per1* (**a**), *Per2* (**b**), *Cry1* (**c**), and *Bmal1* (**d**) mRNA in the SCN of wild-type and *Ptgfr-KO* mice maintained under DD conditions. (**e**) Representative in situ hybridization images. *** *p* < 0.001; significantly different at the corresponding time points (two-way ANOVA with Sidak’s post hoc test; time: F_5,10_ = 35.38, *p* < 0.001; group: F_1,2_ = 37.72, *p* < 0.05; interaction: F_5,10_ = 6.968, *p* < 0.01). The results of the Cosinor analysis are shown in Appendix A.

**Figure 3 ijms-25-01841-f003:**
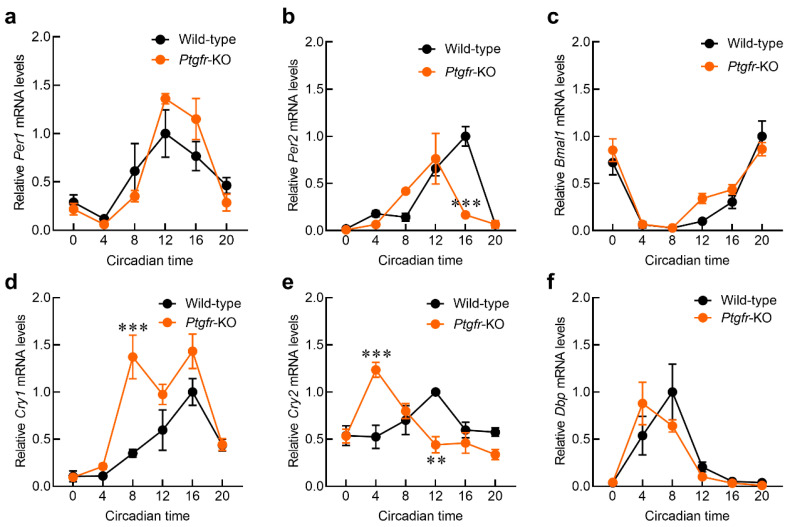
Time-dependent expression of the hepatic clock control gene in *Ptgfr-KO* mice. (**a**–**f**) Temporal expression roles of *Per1* (**a**), *Per2* (**b**), *Bmal1* (**c**), *Cry1* (**d**), *Cry2* (**e**), and *DBP* (**f**) mRNA in the livers of wild-type and *Ptgfr-KO* mice maintained under DD conditions. Values are shown as means with standard error; (*n* = 3). ** *p* < 0.01, *** *p* < 0.001; significant difference from wild-type mice at the corresponding time points (two-way ANOVA with Sidak’s post hoc test). The results of the Cosinor analysis are shown in Appendix A.

**Figure 4 ijms-25-01841-f004:**
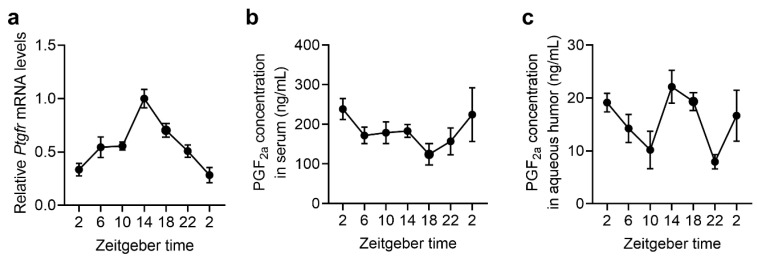
Diurnal variation in the retinal expression of *Ptgfr* mRNA and PGF_2α_ concentration in the sera and aqueous humor of wild-type mice. (**a**) Temporal expression profiles of *Ptgfr* mRNA in wild-type mice. The mean peak values were set at 1.0. Values are shown as means with standard error (*n* = 4–6). Significant time-dependent variations were observed in *Ptgfr* mRNA levels in wild-type mice (*p* < 0.001, one-way ANOVA). (**b**,**c**) Diurnal variation in PGF_2α_ concentrations in the serum (**b**) and the aqueous humor (**c**). Values are shown as means with standard error (*n* = 3–4). Significant time-dependent variations were observed in the PGF_2α_ concentrations in the aqueous humor of mice (one-way ANOVA, serum: *p* = 0.227; aqueous humor: *p* < 0.05). The results of the Cosinor analysis are shown in Appendix A.

**Figure 5 ijms-25-01841-f005:**
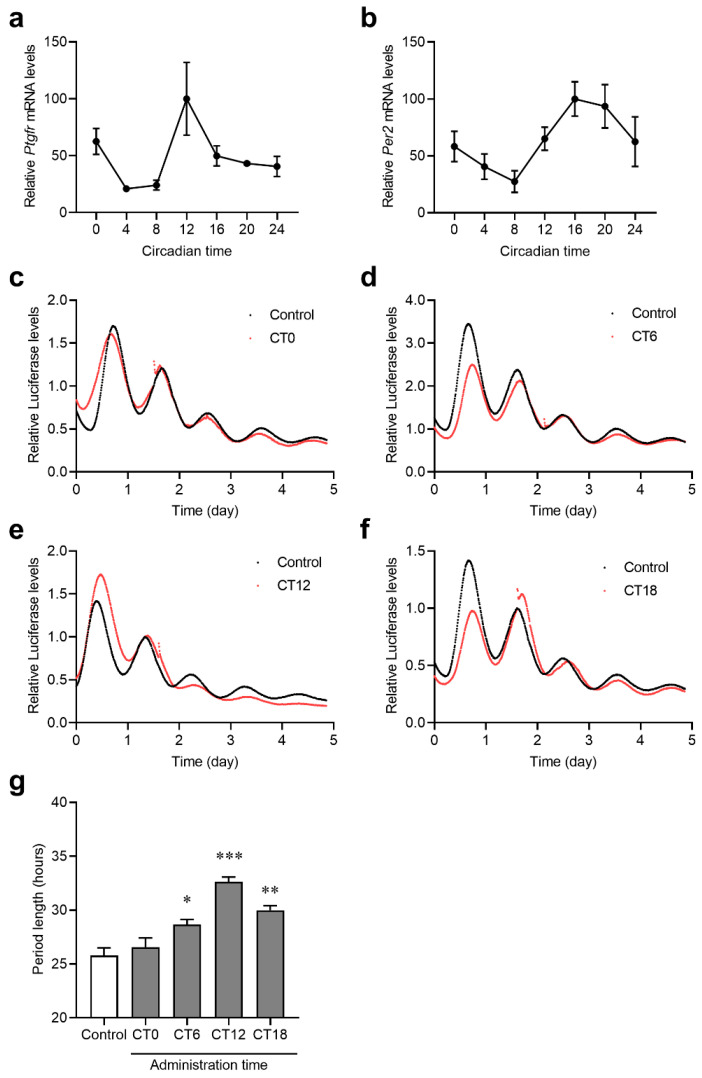
Influence of latanoprost on *Per2* gene transcriptional activity cycle in *Per2*::luc C6 cells. (**a**,**b**) Temporal expression profile of *Ptgfr* and *Per2* mRNA in *Per2*::luc C6 cells after DEX treatment. CT0 represents 30 h after dexamethasone (DEX) addition. Each value represents the mean with standard error (*n* = 3–4). Significant time-dependent variations were observed in *Ptgfr* and *Per2* mRNA expression (one-way ANOVA, *Ptgfr* mRNA: *p* < 0.05; *Per2* mRNA: *p* < 0.05). (**c**–**f**) *Per2*::luc activity variation with the treatment of latanoprost (red) or saline (black) on CT0 (**c**), 6 (**d**), 12 (**e**), and 18 (**f**). Latanoprost was added 1–2 days after DEX synchronization. Each value represents the mean with standard error (*n* = 4–5). (**g**) *Per2*::luc transcription period after latanoprost treatment. Each value represents the mean with standard error; (*n* = 4–5). * *p* < 0.05, ** *p* < 0.01, *** *p* < 0.001; significant difference from saline-treated samples (F_4,19_ = 20.47, *p* < 0.001; one-way ANOVA, Dunnet’s post hoc test). The results of the Cosinor analysis are shown in Appendix A.

## Data Availability

Data from this study are available from the corresponding author upon reasonable request.

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
