# Peer review of "Prostaglandin F2α Affects the Cycle of Clock Gene Expression and Mouse Behavior"

_ijms, 2024, doi:10.3390/ijms25031841_

Round 1

Reviewer 1 Report

Comments and Suggestions for Authors

This manuscript investigates the role of prostaglandins and their receptors, specifically FP, in the regulation of clock genes and biological rhythms. The major issue is that the authors only examined the expression of clock genes in the SCN and liver of FP-KO mice, as well as the rhythms of FP receptor expression and Pgf2a secretion. However, they did not provide sufficient evidence to establish a connection between these factors.

Also, here are several major revisions that need to be addressed before it can be considered for publication.

1.    The notation for gene symbols throughout the article is inconsistent. For example, the genes Per1 and Cry1 are sometimes written in italics and sometimes not. Additionally, the notation for F2a or F2α needs to be standardized.

2.    In the legend for Figure 1, only panel (A) is mentioned. However, in panel (B), it is stated that the wild-type (WT) cycle is approximately 23.6 hours, while the main text mentions 23 hours. Please clarify and ensure consistency.

3. The background section of the article needs to provide an introduction to Pgf2a and its relationship with the FP receptor.

4.    It is unclear why the gene expression analysis in peripheral tissues only focuses on the liver. Please justify this selection and consider including other relevant tissues.

5.    Some of the conclusions drawn in the article are not well-supported. For example, the statement "These results suggest that the prolonged behavioral cycle of FP-KO mice is caused by altered expression of Cry1 mRNA in the SCN" contradicts previous experimental findings that low expression of Cry1 leads to a shorter circadian cycle. Similarly, the conclusion "Because the expression of clock genes in the SCN and liver also showed a significant change in expression pattern during the dark period, Pgf2a signaling is activated during the dark period and may affect the expression rhythm of clock genes" appears premature and requires further experimental validation.

6.    From the lumicycle curve in Figure 5, it is not evident that there is a significant elongation of the circadian cycle at CT12. Please provide clearer evidence or explanation for this claim.

Comments on the Quality of English Language

Overall, the quality of English language in the manuscript is satisfactory.

Reviewer 2 Report

Comments and Suggestions for Authors

The manuscript by Yuya Tsurudome et al., entitled:

“Prostaglandin F2α affects the expression cycle of clock genes and mouse behavior”

reveals interesting findings of role of FP prostaglandin receptor on biological rhythms and clock gene expression.

However, important analysis in results are lacking, including Cosinor analysis, which would clarify numbers and statistical significance of the results. Moreover, English expressions and sentences are confusing, and standard nomenclature in Chronobiological studies is not well used.

Thus, it is strongly advised that the manuscript is thoroughly reviewed by an expert chronobiologist who run Cosinor analysis, complete results, and review sentences and expressions in order to properly describe the circadian variations found in the tested parameters. English expressions should also be checked.

Somme comments follow:

L28: “Prostaglandins are one of the bioactive substances and their receptors affect the expression of clock genes.” You may want to substitute substances by compounds. You may want to substitute “their” by “activation of their receptors”.

L30: “We observed that the cycle was prolonged under DD”. You may want to modify expression, eg. longer circadian rhythm period lengths compared to controls, tau?.

L32: “observed a decrease in the expression amplitude and a change in the expression cycle of the clock gene Cry1, which is related to the cycle”. Sentence is not clear. What happened to Cry1?

L34: A circadian rhythm was also observed in the plasma Pgf2a levels of wild-type mice. You may want to add something like “but not in the WT”, if this is the case?

L35: “Because the addition of Pgf2a agonists at various times in the in vitro experiments altered the amplitude and expression intensity” of what?

57: “The suprachiasmatic nucleus (SCN) in the brain is the center of the clock mechanism”. Center?

58: “Receipt of a light signal that has been cured in the retina.” Receipt? Please, check English.

68: “Although they are known to affect the expression of clock genes.” Please, add references.

72: “We found that the behavioral cycle was prolonged in FP-KO mice.” (phase of activity?)

L82: “WT mice exhibited a behavioral cycle of approximately 23 h (Fig. 1A), similar to that in previous reports [12]. In contrast, FP-KO mice exhibited a longer behavioral cycle (approximately 24 h).” Please, avoid “aproximately”. Please, run Cosinor analysis to characterize the exact circadian period (tau).

94 “FP-KO mice have a prolonged behavioral cycle”. You may want to modify expression, eg. longer circadian rhythm period lengths (tau), compared to controls.

95 “We measured the expression rhythms of Per1, Per2, Bmal1, and Cry1, which form the core of the body clock” (why expression of Clock was not measured?). Please, add where you measured. You may want to check expression “Body clock”.

99 “the Cry1 mRNA expression cycle was prolonged in FP-KO mice” (please explain better).

259: “Slices containing SCN” do you mean suprachiasmatic nucleus? Please include suprachiasmatic nucleus the first time you mention abbreviation of SCN in the text. Please, show at least a couple of representative images of the in situ hybridization analysis of the SCN.

129 “2.4. FP receptor expression rhythms and Pgf2a secretion rhythms in mice” Instead of in mice you may want to write “in retina”.

131 “Circadian rhythms in prostaglandin secretion and prostaglandin receptor expression have also been observe.” Please, include where.

133 “we measured the expression rhythm of FP receptors in the retina.” Please, include by what methodology.

137 “the Pgf2a concentrations in the aqueous humor exhibited the same phase rhythm as the FP mRNA expression rhythm” mRNA expression rhythms where? Please add “in…” at the end of the sentence”

139 “Because the expression of clock genes in the SCN and liver also showed a significant change in expression pattern during the dark period, Pgf2a signaling is activated during the dark period and may affect the expression rhythm of clock genes.”. You may want to rewrite this sentence.

142 “2.5. Effects of FP receptor agonists on Per2 expression rhythm” you may want to add in cell culture.

151: “Figure 4 Diurnal variation in FP receptor expression and PGF2a concentration in the wild-type mice  (A).” Please, include where.

Fig 5F: You may want to check red line in the figure.

173 “the expression cycle of Per2 mRNA was attenuated in the livers of FP-KO mice compared to that of WT” You mean the amplitude? And acrophase was also advanced.

175 “whereas the expression cycle of Cry1 and Cry2 mRNA was augmented.” Please, rewrite the sentence to properly describe the circadian variations.

230 “The mechanisms of biological clock modulation include shift work, irregular meal times, and long light reception times.” You may want to check the sentence, knowing that the main synchronizer is light, but also meals and other time clues may act as zeitgebers.

259 “At six time points, the brain of each mouse was harvested and flash-frozen.” Please, add every 4 hours starting ZT…

276 “Plasma and aqueous humor samples from WT mice were collected at six time points (ZT2, 6, 10, 14, 18, and 22).” In order to fully understand the cycle, a last time point should have been included, which would be done at ZT2+ 24 h. Similar comment for every sampling in the manuscript.

Comments on the Quality of English Language

English expressions and sentences are confusing, and standard nomenclature in Chronobiological studies is not well used. See examples above.

Reviewer 3 Report

Comments and Suggestions for Authors

I have read the article by Tsurudome et al with great interest. The authors investigated the effect of prostaglandin on the molecular clock.

Comments:

·       Abstract. Please, explain all abbreviations in the abstract.

·       Introduction. 2nd paragraph. Please, explain what is CREB.

·       Introduction. Last paragraph. The authors use Prostaglandin F2alpha and FP interchangeably throughout the manuscript. Can they please, explain in more detail with references, if other ligands can also act through this receptor?

·       Methods/Results. I understand the fluid and water intake was ad libitum. However, circadian rhythm strongly affects metabolism and vice versa. Could you please, provide any information on the feeding or weight changes?

·       Discussion. Could you please, explain why did FP knockout have different effect on SCN and liver cells?

·       Discussion. Last paragraph. You may consider citing https://pubmed.ncbi.nlm.nih.gov/37762448/ which is a recent review explaining the role of circadian rhythm in metabolism and inflammation.  

Round 2

Reviewer 2 Report

Comments and Suggestions for Authors

Thank you for your thoughful responses and modifications to the document that clearly improved its quality.

Reviewer 3 Report

Comments and Suggestions for Authors

I am happy with the changes and suggest acceptance